

# Eating habits, lifestyle behaviors and stress during the COVID-19 pandemic quarantine among Peruvian adults

Hellen S. Agurto[1], Ana L. Alcantara-Diaz[2,3], Eduardo Espinet-Coll[4] and Carlos J. Toro-Huamanchumo[3,5]

[1] Universidad Peruana Cayetano Heredia, Lima, Peru
[2] SCIEMVE, Sociedad Científica Veritas, Chiclayo, Peru
[3] Unidad de Investigación Multidisciplinaria, Clínica Avendaño, Lima, Peru
[4] Gastrodex, Hospital Universitario Quiron Dexeus, Barcelona, Spain
[5] Unidad de Investigación para la Generación y Síntesis de Evidencias en Salud, Universidad San Ignacio de Loyola, Lima, Peru

## ABSTRACT

**Background and aims:** The coronavirus disease 2019 (COVID-19) outbreak has led to an unprecedented public health crisis. In Peru, although the quarantine is no longer mandatory, it was during the first months of 2020. To date, no studies have assessed the impact of the COVID-19 on the eating patterns and lifestyle context in the country. We aimed to describe the eating habits, lifestyle behaviors and stress during the COVID-19 pandemic quarantine among Peruvian adults.

**Methods:** We conducted a cross-sectional study. We used an online survey to collect information regarding eating habits, self-perceived stress and sedentary lifestyle among adults over 18 years of age residing in Lima-Peru and who complied with strict home quarantine. We presented our data according to the weight variation of the participants.

**Results:** A total of 686 were finally included in the study. The 82.9% were female, the median BMI was 25.97 kg/m$^2$ (IQR: 23.37–29.41) and 68.2% reported a significant variation in their weight (38.9% increased and 29.3% lost weight). All bad habits were significantly associated with weight gain, except for prolonged fasting. Additionally, a sitting time longer than usual ($p = 0.001$), being in front of a screen for more than five hours in the last week ($p = 0.002$), and most of the stressful scenarios were significantly associated with weight gain.

**Conclusion:** Almost four out of ten participants gained weight during the quarantine. This was associated with unhealthy eating habits, physical inactivity, and stressful scenarios.

Corresponding author
Carlos J. Toro-Huamanchumo, toro2993@hotmail.com

# INTRODUCTION

The ongoing coronavirus disease 2019 (COVID-19) outbreak has led to an unprecedented public health crisis and a global health emergency (*Arshad Ali et al., 2020*). By the end of October 2020, more than 42 million cases and more than 1.1 million deaths had been reported worldwide (*World Health Organization, 2020a*). America is the continent with

the highest rate of reported COVID-19 cases, with USA, Brazil and Peru as the most affected countries (*Acosta, 2020*; *Rodriguez-Morales et al., 2020*).

Due to the rapid widespread and severe public health disruption, the World Health Organization (WHO) recommended various strategies to reduce the COVID-19 transmission. For example, physical distancing, home quarantine, closure of schools, universities and non-essential businesses, among others (*World Health Organization, 2020b*). Regarding quarantine, although it has had a positive impact on reducing the transmission of COVID-19 (*Sen, Karaca-Mandic & Georgiou, 2020*; *Pan et al., 2020*), some studies have reported that it could trigger high levels of anxiety, depressive symptoms and post-traumatic stress disorders (*Brooks et al., 2020*; *Fawaz & Samaha, 2020*; *Guo et al., 2020*). Naturally, this could also predispose to changes in lifestyles and unhealthy nutritional habits (*Papandreou et al., 2020*; *Di Renzo et al., 2020*; *Górnicka et al., 2020*; *Ruiz-Roso et al., 2020b*).

In Peru, the state of emergency became official on March 16 (*Presidencia del Consejo de Ministros, 2010*). Since then, physical distancing measures and home quarantine were promoted, and although the quarantine is no longer mandatory (there are only some restriction hours), it was during the first months. To date, few studies have described some lifestyles during the quarantine in Latin America (*Werneck et al., 2020*; *Ruíz-Roso et al., 2020c*). However, none of them have been conducted in Peru, which is one of the countries with the highest number of cases and deaths due to COVID-19 worldwide (*World Health Organization, 2021*).

This study aimed to describe the eating habits, lifestyle behaviors and stress during the COVID-19 pandemic quarantine among Peruvian adults.

## METHODS

### Study design
We conducted a cross-sectional study in July, during the mandatory quarantine in Peru.

### Study population and context
We included individuals aged 18 years and over who were in Lima at the survey time and with a good precision of the self-reported weight and height (≥8 points on a 0–10 scale). For the latter, participants were asked on a scale of 0 (no accuracy) to 10 (total accuracy) to report how accurate they felt when answering the questions regarding their anthropometric values. Surveys with incomplete information were not considered for analysis.

Lima is the capital of Peru and is the department with the largest population in the country. According to the National Institute of Statistics and Informatics, by 2018, it had a total population of 9 million 320,000 inhabitants(*Instituto Nacional de Estadística e Informática, 2018*).

### Variables and instruments
We developed a self-administered web-based four-section survey. The first section collected sociodemographic and self-reported anthropometric data (age, sex, self-reported

height and weight and precision of the self-report, marital status and self-report of weight variation from the beginning of the quarantine until the moment of the survey). Weight variation was further categorized in "lost weight" (if the participant lost at least 2.5 kg), "weight stable" (no changes or a variation less than 2.5 kg) and "gained weight" (if the participant gained at least 2.5 kg). The second section consisted of an eating habits questionnaire previously validated in a Spanish-speaking population (*Reséndiz Barragán et al., 2015*). This questionnaire had 17 items with Likert-type responses ranging from 0 to 6 (0 = never, 1 = less than once a month, 2 = once a month, 3 = two or three times a month, 4 = once or twice per week, 5 = three or four times a week, and 6 = every day). For the present study, we categorized the responses as: never, one to three times a month, one to four times a week, every day.

For the third and fourth sections, we used some questions from the Spanish versions of the "Last 7 days sedentary behavior questionnaire" (SIT-Q) (*Felez-Nobrega et al., 2019*) and the "Perceived Stress Scale" (PSS) (*Baik et al., 2019*; *Remor, 2006*). Both instruments have been used in different studies during the quarantine in the context of COVID-19 (*Zachary et al., 2020*; *Gallè et al., 2020*; *Iasevoli et al., 2020*). It is important to mention that, since the present study's objective was mainly descriptive, we opted to only select some questions from each questionnaire.

## Procedures

The survey link was distributed using social media (Facebook and other social networking sites) to reach the highest number of participants from all the districts in Lima, Peru. The informed consent was at the beginning of the online (Google® form) survey, including the estimated time needed to complete the survey (15–20 min). If a participant wanted to receive his/her detailed results in a personalized way, he/she was asked to enter his/her e-mail. It is important to mention that all participants voluntarily opted for this option. This also allowed us to check and drop duplicates before the analysis.

## Statistical analysis

We presented the descriptive results for numeric variables as means with their standard deviation (SD) or medians with interquartile range (IQR). Categorical variables were presented as frequencies and percentages. According to the weight variation, each of the eating habits, sedentary behaviors, and stressful scenarios items were compared using the one-way ANOVA or the Kruskal Wallis test as appropriate for continuous variables, and the Chi2 or Fisher exact test for categorical variables. We used STATA v16.0 for our analyses.

## Ethics

The Impacta Institutional Review Board, Lima, Peru (RCEI-17) approved the present study (00110-2020-CE). We did not collect personal data, and participation was voluntary and anonymous. The first page of the survey had the consent form. The participants that agreed had to mark the option "I have read the consent form, and I agree with it. I would

like to start the survey". Only the participants who marked this option were able to continue with the following survey questions.

## RESULTS

A total of 1,031 adults completed the survey, and 686 considered that they could report their weight with an accuracy of eight points or more. The 82.9% were female, 66% were single, and the median age was 31 (IQR: 25-41). The median BMI was 25.97 kg/m$^2$ (IQR: 23.37–29.41) and 68.2% reported a significant variation in their weight (38.9% increased and 29.3% lost weight) from the start of quarantine to the date of the application of the survey.

Tables 1–3 reports absolute and relative frequencies of eating habits, sedentary behavior, stressful scenarios and weight variation during the quarantine. Relative frequencies were calculated by rows variables (Questions). Table 1 shows significant differences in most of the eating habits. All bad habits were significantly associated with weight gain, except for prolonged fasting (that was associated with weight loss, $p = 0.028$). Almost 50% of the participants who reported gaining weight answered that they had snacks between meals ($p = 0.030$) or had sugar cravings every day ($p = 0.001$). Similarly, these participants were the ones who reported having significantly more snacks between meals ($p = 0.002$). Additionally, a sitting time longer than usual ($p = 0.001$), being in front of a screen for more than five hours in the last week ($p = 0.002$), and most of the stressful scenarios were significantly associated with weight gain (Tables 2 and 3). Some of these scenarios included being upset because of something that happened unexpectedly ($p = 0.001$), felt nervous and stressed ($p = 0.001$), found that they could not cope with all the things that they had to do ($p = 0.024$), and felt difficulties were piling up so high that they could not overcome them ($p = 0.020$).

## DISCUSSION

Before describing the eating behaviors during quarantine, it is important to note that it is expected that the availability of food would be restricted during this period. This already makes it difficult to eat healthy foods (*Díez, Bilal & Franco, 2019*; *Bilal et al., 2018*). Also, quarantine itself reduces the possibility of physical activity, promoting a sedentary lifestyle (*Fernández-Sanjurjo et al., 2018*), which can also generate a significant neuropsychiatric burden (*Troyer, Kohn & Hong, 2020*). This has been evidenced by our results, as the participants manifested different stressful situations during the quarantine. This, in turn, may be related to the excessive intake of "comfort foods" (such as pizzas, fried chicken, burgers, French fries, among others) (*Moynihan et al., 2015*), which naturally leads to weight gain. These foods, mainly rich in sugar and carbohydrates, can reduce stress as they stimulate serotonin production with a positive effect on mood (*Lima et al., 2020*). However, this food-craving effect of carbohydrates is proportional to the glycemic index of these "comfort foods", associated with an increased risk of developing obesity and cardiovascular disease, which increase the risk of more severe complications from COVID-19 (*Yannakoulia et al., 2008*).

**Table 1  Eating habits and weight variation during quarantine.**

| Question | Lost weight ($n$ = 201) | Weight stable ($n$ = 218) | Gained weight ($n$ = 267) | $p$ |
|---|---|---|---|---|
| *"How often do you…"* | | | | |
| Snack between meals? | | | | 0.030* |
|    Never | 24 (42.9) | 15 (26.8) | 17 (30.4) | |
|    One to three times a month | 32 (33.3) | 35 (36.5) | 29 (30.2) | |
|    One to four times a week | 85 (27.6) | 106 (34.4) | 117 (38.0) | |
|    Everyday | 60 (26.6) | 62 (27.4) | 104 (46.0) | |
| Have long fasting periods? | | | | 0.028* |
|    Never | 99 (27.1) | 106 (29.0) | 160 (43.8) | |
|    One to three times a month | 43 (29.3) | 56 (38.1) | 48 (32.7) | |
|    One to four times a week | 40 (29.9) | 45 (33.6) | 49 (36.6) | |
|    Everyday | 19 (47.5) | 11 (27.5) | 10 (25.0) | |
| Eat until you feel uncomfortable? | | | | 0.001* |
|    Never | 79 (30.7) | 98 (38.1) | 80 (31.1) | |
|    One to three times a month | 90 (31.4) | 97 (33.8) | 100 (34.8) | |
|    One to four times a week | 28 (23.7) | 19 (16.1) | 71 (60.2) | |
|    Everyday | 4 (16.7) | 4 (16.7) | 16 (66.7) | |
| Eat without feeling physical hunger? | | | | 0.001* |
|    Never | 56 (30.0) | 77 (41.2) | 54 (28.9) | |
|    One to three times a month | 81 (31.9) | 89 (35.0) | 84 (33.1) | |
|    One to four times a week | 55 (29.7) | 41 (22.2) | 89 (48.1) | |
|    Everyday | 9 (15.0) | 11 (18.3) | 40 (66.7) | |
| Feel guilty or sad after eating? | | | | 0.001* |
|    Never | 72 (31.4) | 103 (45.0) | 54 (23.6) | |
|    One to three times a month | 83 (34.3) | 78 (32.2) | 81 (33.5) | |
|    One to four times a week | 32 (26.2) | 23 (18.8) | 67 (54.9) | |
|    Everyday | 14 (15.1) | 14 (15.1) | 85 (34.3) | |
| Stressed by the way you eat? | | | | 0.001* |
|    Never | 85 (34.3) | 111 (44.8) | 52 (21.0) | |
|    One to three times a month | 54 (27.6) | 71 (36.2) | 71 (36.2) | |
|    One to four times a week | 43 (34.1) | 20 (15.9) | 63 (50.0) | |
|    Everyday | 19 (16.4) | 16 (13.8) | 81 (69.8) | |
| Drink sodas, processed juices or shakes? | | | | 0.001* |
|    Never | 47 (32.2) | 57 (39.0) | 42 (28.8) | |
|    One to three times a month | 119 (32.0) | 115 (30.9) | 138 (37.1) | |
|    One to four times a week | 31 (20.7) | 39 (26.0) | 80 (53.3) | |
|    Everyday | 4 (22.2) | 7 (38.9) | 7 (38.9) | |
| Drink water? | | | | 0.001* |
|    Never | 3 (15.8) | 3 (15.8) | 13 (68.4) | |
|    One to three times a month | 9 (34.6) | 3 (11.5) | 14 (53.9) | |
|    One to four times a week | 26 (21.9) | 33 (27.7) | 60 (50.4) | |
|    Everyday | 163 (31.2) | 179 (34.3) | 180 (34.5) | |
| Have sugar cravings? | | | | 0.001* |
|    Never | 11 (36.7) | 11 (36.7) | 8 (26.7) | |
|    One to three times a month | 75 (36.6) | 70 (34.2) | 60 (29.3) | |
|    One to four times a week | 74 (28.7) | 90 (34.9) | 94 (36.4) | |
|    Everyday | 41 (21.24) | 47 (24.4) | 105 (54.4) | |

*(Continued)*

| Question | Lost weight (n = 201) | Weight stable (n = 218) | Gained weight (n = 267) | p |
|---|---|---|---|---|
| Have salt cravings? | | | | 0.001* |
| Never | 20 (31.8) | 26 (41.3) | 17 (27.0) | |
| One to three times a month | 86 (34.4) | 88 (35.2) | 76 (30.4) | |
| One to four times a week | 65 (24.9) | 75 (28.7) | 121 (46.4) | |
| Everyday | 30 (26.8) | 29 (25.9) | 53 (47.3) | |
| Have cravings for fatty foods? | | | | 0.001* |
| Never | 21 (25.92) | 39 (48.2) | 21 (25.9) | |
| One to three times a month | 123 (33.8) | 125 (34.3) | 116 (31.9) | |
| One to four times a week | 47 (24.1) | 44 (22.6) | 104 (53.3) | |
| Everyday | 10 (21.7) | 10 (21.7) | 26 (56.5) | |
| Drink natural juices? | | | | 0.007* |
| Never | 30 (43.5) | 16 (23.2) | 23 (33.3) | |
| One to three times a month | 72 (28.8) | 82 (32.8) | 96 (38.4) | |
| One to four times a week | 71 (29.3) | 66 (27.3) | 105 (43.4) | |
| Everyday | 28 (22.4) | 54 (43.2) | 43 (34.4) | |
| Leave the plate "empty" when you finish eating? | | | | 0.538* |
| Never | 10 (38.5) | 10 (38.5) | 6 (23.1) | |
| One to three times a month | 9 (25.0) | 10 (27.8) | 17 (47.2) | |
| One to four times a week | 41 (29.1) | 40 (28.4) | 60 (42.6) | |
| Everyday | 141 (29.2) | 158 (32.7) | 184 (38.1) | |
| Have breakfast in the week? | | | | 0.016* |
| One to two days | 24 (39.3) | 18 (29.5) | 19 (31.2) | |
| Three to five days | 32 (25.4) | 30 (23.8) | 64 (50.8) | |
| Six to seven days | 145 (29.1) | 170 (34.1) | 184 (36.9) | |
| Have lunch in the week? | | | | 0.756* |
| One to two days | 5 (25.0) | 6 (30.0) | 9 (45.0) | |
| Three to five days | 10 (23.8) | 12 (28.6) | 20 (47.6) | |
| Six to seven days | 186 (29.8) | 200 (32.1) | 238 (38.1) | |
| Have dinner in the week? | | | | 0.016* |
| One to two days | 29 (40.9) | 19 (26.8) | 23 (32.4) | |
| Three to five days | 37 (21.8) | 52 (30.6 ) | 81 (47.7) | |
| Six to seven days | 135 (30.3) | 147 (33.0) | 163 (36.6) | |
| *"How many…"* | | | | |
| Main meals do you have per day? | 3 (2-3) | 3 (2-3) | 3 (2-3) | 0.198[†] |
| Snacks between meals do you have per day? | 1.30 (0.88) | 1.25 (0.79) | 1.51 (0.82) | 0.002[††] |
| *D. you…* | | | | |
| Skip meals to take care of your figure? | | | | 0.001* |
| No | 53 (21.0) | 88 (34.8) | 112 (44.3) | |
| Yes | 148 (34.2) | 130 (30.02) | 155 (35.8) | |

**Notes:**

* Chi2 test.
[†] Kruskal Wallis.
[††] ANOVA.

COVID-19 has had a negative psychological impact worldwide, not only due to the risk of infection but also due to the different measures implemented to contain the outbreak spread (*Guo et al., 2020*; *Lal et al., 2020*). During the COVID-19 outbreak, several

**Table 2 Sedentary behavior and weight variation during quarantine.**

| Question: "In the last week, how long…" | Lost weight (n = 201) | Weight stable (n = 218) | Gained weight (n = 267) | p |
|---|---|---|---|---|
| Have you been sitting or lying down? | | | | 0.001* |
| Less than normal | 36 (43.9) | 25 (30.5) | 21 (25.6) | |
| About the same | 45 (35.7) | 50 (39.7) | 31 (24.6) | |
| More than usual | 120 (25.1) | 143 (29.9) | 215 (45.0) | |
| Did you sit for breakfast, lunch, or dinner? | | | | 0.940* |
| <30 min | 129 (29.5) | 142 (32.5) | 166 (38.0) | |
| 30–60 min | 51 (30.0) | 51 (30.0) | 68 (40.0) | |
| >1 h | 21 (26.6) | 25 (31.7) | 33 (41.8) | |
| Did you sit or lie down in front of a screen? | | | | 0.002* |
| <60 min | 51 (33.1) | 54 (35.1) | 49 (31.8) | |
| 1–3 h | 60 (35.1) | 55 (32.2) | 56 (32.8) | |
| 3–5 h | 30 (29.4) | 38 (37.3) | 34 (33.3) | |
| >5 h | 60 (23.2) | 71 (27.4) | 128 (49.4) | |
| Did you sit while reading a book? | | | | 0.469* |
| <60 min | 171 (29.7) | 183 (31.8) | 221 (38.4) | |
| 1–3 h | 26 (28.9) | 25 (27.8) | 39 (43.3) | |
| 3–5 h | 4 (19.1) | 10 (47.6) | 7 (33.3) | |
| Did you sit while playing cards or solving puzzles? | | | | 0.244[†] |
| <60 min | 177 (29.1) | 187 (30.8) | 244 (40.1) | |
| 1–3 h | 18 (28.6) | 27 (42.9) | 18 (28.6) | |
| 3–5 h | 6 (40.0) | 4 (26.7) | 5 (33.3) | |
| Did you sit while listening to music? | | | | 0.333* |
| <60 min | 134 (28.4) | 157 (33.3) | 181 (38.4) | |
| 1–3 h | 47 (34.8) | 39 (28.9) | 49 (36.3) | |
| 3–5 h | 20 (25.3) | 22 (27.9) | 37 (46.8) | |

**Notes:**
[*] Chi2 test.
[†] Fisher exact test.

studies have reported an increase in the prevalence of eating disorders (*Cooper et al., 2020*; *Baenas et al., 2020*; *Phillipou et al., 2020*). Similarly, lower psychological health has been associated with higher body shape and weight concerns (*Haddad et al., 2020*). In this sense, this quarantine can be defined as an unprecedented stressful event that has negatively affected individuals' eating patterns (*Bin Zarah, Enriquez-Marulanda & Andrade, 2020*). In our study, many participants frequently reported unhealthy eating habits, such as eating until feeling uncomfortable, eating without feeling physical hunger, and feeling guilty or sad after eating. Our results agree with current evidence that suggests a strong relationship between unhealthy eating behaviors and stress, anxiety and other mental disorders (*Yau & Potenza, 2013*).

Regular physical activity can be beneficial not only for weight loss but also for strengthening the immune system (*Zheng et al., 2015*). In fact, low–moderate exercise has proven to be beneficial for the innate immune response against respiratory infections

**Table 3 Stressful scenarios and weight variation during quarantine.**

| Question: "In the last month, how often have you…" | Lost weight (n = 201) | Weight stable (n = 218) | Gained weight (n = 267) | p |
|---|---|---|---|---|
| Been upset because of something that happened unexpectedly | | | | 0.001* |
| Never or almost never | 57 (32.8) | 64 (36.8) | 53 (30.5) | |
| Sometimes | 90 (28.0) | 113 (35.1) | 119 (37.0) | |
| Fairly often or very often | 54 (28.4) | 41 (21.6) | 95 (50.0) | |
| You felt that you were unable to control the important things in your life? | | | | 0.051* |
| Never or almost never | 83 (31.9) | 89 (34.2) | 88 (33.9) | |
| Sometimes | 78 (29.1) | 88 (32.8) | 102 (38.1) | |
| Fairly often or very often | 40 (25.3) | 41 (26.0) | 77 (48.7) | |
| Felt nervous and "stressed"? | | | | 0.001* |
| Never or almost never | 39 (37.5) | 32 (30.8) | 33 (31.7) | |
| Sometimes | 74 (28.2) | 102 (38.9) | 86 (32.8) | |
| Fairly often or very often | 88 (27.5) | 84 (26.3) | 148 (46.3) | |
| Felt confident about your ability to handle your personal problems? | | | | 0.263* |
| Never or almost never | 14 (26.4) | 19 (35.9) | 20 (37.7) | |
| Sometimes | 51 (25.1) | 61 (30.1) | 91 (44.8) | |
| Fairly often or very often | 136 (31.6) | 138 (32.1) | 156 (36.3) | |
| Felt that things were going your way? | | | | 0.001* |
| Never or almost never | 13(33.3) | 9 (23.1) | 17 (43.6) | |
| Sometimes | 49 (20.6) | 75 (31.5) | 114 (47.9) | |
| Fairly often or very often | 139 (34.0) | 134 (32.8) | 136 (33.3) | |
| Found that you could not cope with all the things that you had to do? | | | | 0.024* |
| Never or almost never | 87 (34.1) | 81 (31.8) | 87 (34.1) | |
| Sometimes | 85 (27.8) | 104 (34.0) | 117 (38.2) | |
| Fairly often or very often | 29 (23.2) | 33 (26.4) | 63 (50.4) | |
| Been able to control irritations in your life? | | | | 0.034* |
| Never or almost never | 3 (14.3) | 9 (42.9) | 9 (42.9) | |
| Sometimes | 44 (26.7) | 42 (25.5) | 79 (47.9) | |
| Fairly often or very often | 154 (30.8) | 167 (33.4) | 179 (35.8) | |
| Felt that you were on top of things? | | | | 0.056* |
| Never or almost never | 13 (21.5) | 24 (36.9) | 27 (41.5) | |
| Sometimes | 60 (24.8) | 75 (31.0) | 107 (44.2) | |
| Fairly often or very often | 127 (33.5) | 119 (31.4) | 133 (35.1) | |
| Felt difficulties were piling up so high that you could not overcome them? | | | | 0.020* |
| Never or almost never | 108 (32.2) | 115 (34.3) | 112 (33.4) | |
| Sometimes | 69 (27.4) | 79 (31.4) | 104 (41.3) | |
| Fairly often or very often | 24 (24.2) | 24 (24.2) | 51 (51.5) | |

Notes:
* Chi2 test.

(*Matricardi, Dal Negro & Nisini, 2020*) and could improve some clinical conditions related with severe COVID-19 (*Dwyer et al., 2020*). However, this has been limited in some cases due to the closure of gyms and public open spaces. We found a high frequency of sedentary lifestyles and too much time in front of screens during the quarantine. Similar results have been reported in other studies worldwide (*Meyer et al., 2020*; *Zheng et al., 2020*; *Ruiz-Roso et al., 2020a*). We also found that this sedentary behavior was related to weight gain among the study participants.

Our study had some limitations. First, the weight variation was self-reported. However, we do not consider this as a continuous variable but rather as an ordinal scale variable. Additionally, participants were asked -on a scale of 1 to 10- to report how accurate they felt they could be answering this question. Only the reliable answers (defined as a score ≥8) were chosen. Second, the study was conducted using an online survey. Thus, the population included was the one that responded on social networks or via e-mail. This could limit the extrapolation of our results to the adult population that has access to social media platforms.

Some strengths should also be highlighted. The current study provides valuable information on eating habits and lifestyle behaviors in the context of an unprecedented event worldwide. Moreover, it is the first published study that have addressed this topic in Peru, which is one of the countries with the highest number of cases and deaths due to COVID-19 worldwide. Our results may be useful for implementing public policies to promote healthy lifestyles during the pandemic. In addition, our study provides insight for future research to implement and evaluate different coping strategies to avoid comorbidities associated with weight gain, especially for future circumstances that will again require self-quarantine.

In conclusion, almost 4 out of 10 participants reported an increase of 2.5–5 kg in their weight. This was related to some unhealthy eating behaviors and a sedentary lifestyle. The awareness of these factors could be an opportunity to promote nutrition and physical activity programs across the country, especially since most of them are potentially modifiable. Additionally, we recommend implementing community-based strategies to promote coping skills and support resilience during the COVID-19 pandemic.

## ACKNOWLEDGEMENTS

To Marilyn Espantoso and Erick Piskulich for their support in data collection.

### Funding

The authors received no funding for this work.

### Competing Interests

The authors declare that they have no competing interests.

## Author Contributions

- Hellen S. Agurto conceived and designed the experiments, performed the experiments, authored or reviewed drafts of the paper, and approved the final draft.
- Ana L. Alcantara-Diaz performed the experiments, prepared figures and/or tables, authored or reviewed drafts of the paper, and approved the final draft.
- Eduardo Espinet-Coll performed the experiments, authored or reviewed drafts of the paper, and approved the final draft.
- Carlos J. Toro-Huamanchumo performed the experiments, analyzed the data, prepared figures and/or tables, authored or reviewed drafts of the paper, and approved the final draft.

## Human Ethics

The following information was supplied relating to ethical approvals (i.e., approving body and any reference numbers):

Impacta Institutional Review Board (Lima, Peru) approved this research (00110-2020-CE).

## Ethics

The following information was supplied relating to ethical approvals (i.e., approving body and any reference numbers):

The Impacta Institutional Review Board, Lima, Peru (RCEI-17) approved the present study (approval 00110-2020-CE).

## Data Availability

The data and codebook are available in the Supplemental File.

## Supplemental Information

Supplemental information for this article can be found online at http://dx.doi.org/10.7717/peerj.11431#supplemental-information.

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
