# Peer review of "Eating habits, lifestyle behaviors and stress during the COVID-19 pandemic quarantine among Peruvian adults"

_PeerJ, doi:10.7717/peerj.11431_

## Round 0.1 · original submission · Major Revisions

Please make sure to include the Ethical Committee approval number, the exclusion criteria, the strengths of the study and the references suggested by the reviewers.

Reviewer 1 ·

Basic reporting

No comment.

Experimental design

No comment.

Validity of the findings

No comment.

Additional comments

General comment:
The authors have studied the impact of COVID-19 pandemic quarantine in the eating habits, lifestyle behaviors and stress of Peruvian adults. Although this question has been addressed in some articles (reference 8-13 of the manuscript), none of them have been conducted in Peru, which is one of the countries with the highest number of cases and deaths due to COVID-19 worldwide. In this context I consider this a clinically relevant question to be considered for publication. However, current version of the article needs some revisions to be addressed before this manuscript is suitable for publication.

Major revisions:
1. In line 82: the authors explains that individuals aged 18 years and over who were in Lima at the survey time and with a good precision of the self-reported weight and height (≥8 points on a 0-10 scale). Please specify survey time window (start date-end date) Please, add an explanation of the good precision criteria [i.e., Participants were asked on a scale of 1 (no accuracy) to 10 (total accuracy) to report how accurate they felt when answering this question].

2. In line 115: the authors explain that eating habits scores were analyzed according to weight variation. Furthermore, weight variation was treated as a qualitative variable. Please define the calculation of this variable.

3. I consider that your results deserve more explanation. For example, in line 134, I suggest that the authors specify what they consider bad habits (i.e., almost 50% of the participants who gained weight answered that they had snacks between meals every day, p- value <0.03). Furthermore, in line 137 the authors could specify stressful scenarios results and Table 2-3.

4. The PSS yields a total score that describes overall perceived stress. The authors used some questions from the Spanish versions of the SIT-Q and PSS. I suggest that the authors could consider the calculation of a total score for their questions. This Score could be used to calculate the impact of weight loss, adjusted by other variables such as sex in a regression model.

5. In line 147, the authors explain that “comfort foods” leads to weight gain. Please define comfort foods in your country (i.e., comfort foods such as …, …, etc.).

6. I am concerned about the control of individuals repetition in the current data. Some persons could have answered the questionnaire several times. Have the authors controlled the repetition of the population? If this problem was controlled, please add an explanation in material and methods. If not please add this problem as a limitation.

Minor revisions:
1) In line 64-65. The authors quoted some literature about the impact of COVID-19 quarantine on psychological parameters (reference 8-10) and nutritional habits (11-13). However, some important publications in other populations are not referenced (i.e., PMID: 32759636, 33296868). Please consider adding them.

2) In line 100-101: Please add Spanish validation reference for SIT-Q and PSS questionnaires.

3) Please add some explanation to tables 1-3. For example, Table 1 reports absolute and relative frequency of eating habits and weight variation during quarantine. Relative frequencies were calculated by rows variables (Questions), etc.

Reviewer 2 ·

Basic reporting

This is a relevant study with important information about lifestyle during the pandemic period.

Experimental design

In methods:-
- I suggest that the article be revised, according to the items contained in the Strobe checklist.
-In the topic Study population and context, the authors said that they included individuals aged 18 years and over who were in Lima at the survey time and with a good precision of the self-reported weight and height (≥8 points on a 0-10 scale). What is a good precision? Is it a questionnaire? It may be explained before.
-Exclusion criteria are missing. It is important to add it.
- In the ethics session, it is important to include the opinion (number) of the ethics committee involved.
-The discussion can be better elaborated because it discusses in a superficial way.
-I suggest including the strengths of the study at the end.
-References about teenagers, but which can be included in the discussion:Ruiz-Roso, M.B.; de Carvalho Padilha, P.; Mantilla-Escalante, D.C.; Ulloa, N.; Brun, P.; Acevedo-Correa, D.; Arantes Ferreira Peres, W.; Martorell, M.; Aires, M.T.; de Oliveira Cardoso, L.; Carrasco-Marín, F.; Paternina-Sierra, K.; Rodriguez-Meza, J.E.; Montero, P.M.; Bernabè, G.; Pauletto, A.; Taci, X.; Visioli, F.; Dávalos, A. Covid-19 Confinement and Changes of Adolescent’s Dietary Trends in Italy, Spain, Chile, Colombia and Brazil. Nutrients 2020, 12, 1807. https://doi.org/10.3390/nu12061807

Validity of the findings

No comment.

---

## Round 0.2 · accepted · Accept

The authors have modified the manuscript as suggested.

Reviewer 1 ·

Basic reporting

no comment

Experimental design

no comment

Validity of the findings

no comment

Additional comments

The authors have studied the impact of COVID-19 pandemic quarantine in the eating habits, lifestyle behaviors and stress of Peruvian adults. Although this question has been addressed in some articles, none of them have been conducted in Peru, which is one of the countries with the highest number of cases and deaths due to COVID-19 worldwide. The authors have made the corrections requested or clarified the reason for not doing some of the recommendations. In this context, I find this article suitable to publish.